# pH-Responsible Doxorubicin-Loaded Fe_3_O_4_@CaCO_3_ Nanocomposites for Cancer Treatment

**DOI:** 10.3390/pharmaceutics15030771

**Published:** 2023-02-26

**Authors:** Victoriya Popova, Yuliya Poletaeva, Alexey Chubarov, Elena Dmitrienko

**Affiliations:** Institute of Chemical Biology and Fundamental Medicine, SB RAS, 8 Lavrentiev Avenue, 630090 Novosibirsk, Russia

**Keywords:** iron oxide and calcium carbonate nanoparticles, Fe_3_O_4_@CaCO_3_, nanocomposites, magnetic nanocomposites, drug delivery, doxorubicin, human serum albumin, pH-sensitive drug release, cancer treatment, toxicity

## Abstract

A magnetic nanocomposite (MNC) is an integrated nanoplatform that combines a set of functions of two types of materials. A successful combination can give rise to a completely new material with unique physical, chemical, and biological properties. The magnetic core of MNC provides the possibility of magnetic resonance or magnetic particle imaging, magnetic field-influenced targeted delivery, hyperthermia, and other outstanding applications. Recently, MNC gained attention for external magnetic field-guided specific delivery to cancer tissue. Further, drug loading enhancement, construction stability, and biocompatibility improvement may lead to high progress in the area. Herein, the novel method for nanoscale Fe_3_O_4_@CaCO_3_ composites synthesis was proposed. For the procedure, oleic acid-modified Fe_3_O_4_ nanoparticles were coated with porous CaCO_3_ using an ion coprecipitation technique. PEG-2000, Tween 20, and DMEM cell media was successfully used as a stabilization agent and template for Fe_3_O_4_@CaCO_3_ synthesis. Transmission electron microscopy (TEM), Fourier transform infrared (FTIR) spectroscopy, and dynamic light scattering (DLS) data were used for the Fe_3_O_4_@CaCO_3_ MNC’s characterization. To improve the nanocomposite properties, the concentration of the magnetic core was varied, yielding optimal size, polydispersity, and aggregation ability. The resulting Fe_3_O_4_@CaCO_3_ had a size of 135 nm with narrow size distributions, which is suitable for biomedical applications. The stability experiment in various pH, cell media, and fetal bovine serum was also evaluated. The material showed low cytotoxicity and high biocompatibility. An excellent anticancer drug doxorubicin (DOX) loading of up to 1900 µg/mg (DOX/MNC) was demonstrated. The Fe_3_O_4_@CaCO_3_/DOX displayed high stability at neutral pH and efficient acid-responsive drug release. The series of DOX-loaded Fe_3_O_4_@CaCO_3_ MNCs indicated effective inhibition of Hela and MCF-7 cell lines, and the IC 50 values were calculated. Moreover, 1.5 μg of the DOX-loaded Fe_3_O_4_@CaCO_3_ nanocomposite is sufficient to inhibit 50% of Hela cells, which shows a high prospect for cancer treatment. The stability experiments for DOX-loaded Fe_3_O_4_@CaCO_3_ in human serum albumin solution indicated the drug release due to the formation of a protein corona. The presented experiment showed the “pitfalls” of DOX-loaded nanocomposites and provided step-by-step guidance on efficient, smart, anticancer nanoconstruction fabrication. Thus, the Fe_3_O_4_@CaCO_3_ nanoplatform exhibits good performance in the cancer treatment area.

## 1. Introduction

Magnetic nanocomposites (MNCs) combine the properties of magnetic nanoparticles (MNPs) and a second material, yielding a broad range of properties of the two phases. Moreover, such incorporation gives a new unique feature, providing widespread applications. The building blocks for MNCs may be organic molecules or polymers, inorganic substances, and bioinspired [1,2,3].

MNCs have widespread applications in drug delivery, magnetic field-influenced transport, magnetic resonance imaging, hyperthermia therapy, theranostics, magnetic separation, and biosensor areas [4,5,6,7,8,9,10,11,12,13,14,15,16,17,18,19,20,21,22,23,24,25,26,27,28,29]. NC is a promising tool for external magnetic field target-specific delivery of anticancer agents [1,21,22,30]. The surfaces of MNPs for in vivo applications should be modified with a highly biocompatible material to acquire excellent stability, good solubility, and low toxicity [1]. The good magnetic core surface protection and low interaction with a solvent primarily lead to high biocompatibility of MNPs [2,3,31,32,33,34]. The core–shell structure of MNCs is generally well tolerated in vivo [25,35,36,37,38]. It provides tunable properties, easy surface functionalization, and low toxicity [25,35,36,37,38].

Calcium carbonate (CaCO_3_) is a well-known mineral applied in various biocompatible materials [39,40,41,42,43,44]. CaCO_3_ is the cheapest inorganic coating with a porous structure, pH-sensitive drug loading and release, and high stability [40,41,42,45]. Recently, we have fabricated nanoscale porous, anticancer drug-loaded CaCO_3_ that showed excellent A549 cell inhibition [41]. Moreover, CaCO_3_ nanoparticles exhibit weak acid decomposition, which can be found in the cancer microenvironment and endosomal compartment, facilitating drug release in tumors [39,40,41]. Hence, the magnetic core/inorganic shell hybrid nanocomposite possesses remarkable priority over other MNCs. Recently, Fe_3_O_4_@CaCO_3_ nanocomposites were designed for metal ions, dyes, magnetic cell separation, enzyme immobilization, and drug adsorption [46,47,48,49,50,51,52,53,54,55,56]. However, only few works show the possible therapeutic applications of Fe_3_O_4_@CaCO_3_ [50]. Despite the advantages, Fe_3_O_4_@CaCO_3_ production is a complex and less-reproducible task. Moreover, the previously synthesized nanocomposites larger than 200 nm form an aggregate. The lower nanoparticle sizes are more suitable for therapeutic applications.

Doxorubicin (DOX) is an outstanding anticancer drug [57,58,59]. The DOX loading on nanoparticles’ surface serves as a prospective system that reduces side effects and avoids drug resistance problems [41,58,59,60,61,62,63,64,65]. Therefore, it is essential to develop new pH-stimuli-responsive DOX-released multifunctional constructions based on MNCs [66,67,68]. Moreover, for further in vivo studies, the excellent capacity of DOX is required. The DOX-loaded MNCs should be stable at a plasma pH of ~7.4 and enable efficient drug release in cancer tissue (pH of 5–5.5) [45,64,66,69,70,71].

Herein, we reported a novel reproducible synthesis of inorganic Fe_3_O_4_@CaCO_3_ nanocomposites with a size of less than 200 nm for biomedical applications. To improve the nanocomposite properties, the concentration of the magnetic core was varied, yielding optimal size, polydispersity, and aggregation ability while maintaining the magnetic properties. The prepared MNCs morphology and aggregation were analyzed using transmission electron microscopy (TEM), Fourier transform infrared (FTIR) spectroscopy, and dynamic light scattering (DLS). The stability experiments of Fe_3_O_4_@CaCO_3_ nanocomposites in acetate buffer (pH 5.0, 7.0), PBS buffer (pH 7.4), DMEM cell media, and 10% fetal bovine serum (FBS) were studied. To serve as a drug carrier, the DOX loading capacity and pH-dependent release kinetic profile were investigated. For this, a series of DOX-loaded Fe_3_O_4_@CaCO_3_ was synthesized. This method provides a high DOX capacity of up to 1900 µg/mg (DOX/MNPs) with 34% loading efficiency or a low capacity of 25 µg/mg with 80% efficiency. The Fe_3_O_4_@CaCO_3_/DOX nanocomposites indicate high pH stability in neutral pH and efficient drug release in acidic media. For Fe_3_O_4_@CaCO_3_/DOX series, the cytotoxicity on HeLa and MCF-7 cell lines was tested. The systemic experimental results per DOX concentration or nanoparticle amount allowed the proof-of-concept of nanocomposite contribution to drug-resistance cancer treatment. The stability experiments with human serum albumin solution in physiological concentration showed the “pitfalls” of DOX-loaded nanocomposites and provided step-by-step guidance on efficient, smart, anticancer nanoconstruction fabrication. We believe that the presented work will bring new quality to the targeted drug delivery area.

## 2. Materials and Methods

### 2.1. Materials

FeCl_2_∙4H_2_O (97–102%), FeCl_3_∙6H_2_O (97–102%), and sodium acetate (99.9%) were obtained from PanReac AppliChem (Barcelona, Spain). Sodium bicarbonate (≥99.7%), calcium chloride, magnesium chloride, oleic acid, boric acid, phosphate-buffered saline (PBS), and Tween 20 were purchased from Sigma (St. Louis, MO, USA). Polyethylene glycol 2000 was obtained from Carl Roth (Karlsruhe, Germany). Doxorubicin was acquired from Ferein (Moscow, Russia). 3-(4,5-Dimethyl-2-thiazolyl)-2,5-diphenyl-2H-tetrazolium bromide (MTT) was purchased from Panreac Química (Barcelona, Spain). Fetal bovine serum (FBS), DMEM (Dulbecco’s modified Eagle medium), GlutaMax, and antimycotic antibiotic solution were obtained from GIBCO, Life Technologies (Carlsbad, CA, USA). Human serum albumin (HSA) was purchased from Renal Laborvegyszer Kereskedelmi Kft. (Budapest, Hungary). Deionized water (Milli-Q) was used to prepare solutions.

### 2.2. Characterization of NPs

Dynamic light scattering (DLS) and zeta potential (ζ-potential) measurements were carried out on Malvern Zetasizer Nano (Malvern Instrument Ltd., Worcestershire, UK). For DLS studies, Fe_3_O_4_@CaCO_3_ was diluted in deionized water to a concentration of 100 μg/mL. Transmission electron microscopy (TEM) images were obtained on a JEM-1400 (Jeol, Tokyo, Japan). Images were captured using a side-mounted Veleta digital camera (EM SIS, Muenster, Germany). UV-vis spectra were recorded on a UV-2100 spectrophotometer (Shimadzu, Kyoto, Japan) and microplate reader Clariostar (BMG, Ortenberg, Germany). FTIR spectra were measured on a 640-IR FT-IR spectrometer (Varian, MA, USA) from 4000 to 400 cm^−1^ at room temperature accompanied with a KBr pellet.

### 2.3. Fe_3_O_4_@CaCO_3_ Nanocomposite Synthesis

The synthesis of oleic acid-coated Fe_3_O_4_ nanoparticles was adapted from Kovrigina et al. and Wang et al. [72,73]. Briefly, the 0.28 g FeCl_3_∙6H_2_O (1 mmol) and 0.1 g FeCl_2_∙4H_2_O (0.5 mmol) were dissolved in 10 mL of HCl (1 M) and heated at 85 °C. Afterward, 95 µL of oleic acid (0.3 mmol) in acetone was added dropwise and stirred (750 rpm) at 85 °C for 5 min. After the incubation, 2 mL of NaOH (8 M) was added up to pH 11.0 under stirring (750 rpm) at 85 °C for 30 min. The mixture was cooled to room temperature. Afterward, 9 mL of HCl (1 M) was added up to pH 2.0. The obtained magnetite nanoparticles were collected using a magnet and washed with 1.0 mL of acetone (three times), and 1.0 mL of deionized water (three times). The nanoparticles were redispersed in deionized water and stored at room temperature.

Fe_3_O_4_@CaCO_3_ nanocomposites were adapted from Popova et al. [41]. Briefly, 0.45–4.5 mg Fe_3_O_4_ and 0.1 g polyethylene glycol 2000 were dissolved in 0.8 mL of deionized water. To the resulting mixture, 0.1 mL of sodium bicarbonate (1 M), 0.1 mL of Tween 20 (20 vol. %), and 0.1 mL of DMEM were added. The 0.1 mL solution, which contains 0.010 mL of calcium chloride (0.1 M), 0.010 mL of magnesium chloride (0.1 M), and 0.010 mL of DMEM, was added slowly under sonification in an ultrasonic bath for 2 min. The mixture was stirred at 25 °C for 20 min (750 rpm). Finally, Fe_3_O_4_@CaCO_3_ was collected using a magnet and redispersed in deionized water. The brown magnetic suspension was kept in deionized water at 23−25 °C.

### 2.4. Fe_3_O_4_@CaCO_3_ Stability

The stability of 100 μg Fe_3_O_4_@CaCO_3_ was analyzed in 1 mL of 100 mM acetate buffer (pH from 5.0 to 6.0), 10 mM PBS (pH 7.4), DMEM, and 10% fetal bovine serum at 25 °C under stirring (750 rpm). At various time points, the aliquot of the solution was collected, followed by re-mixing. The aliquots were analyzed using DLS.

### 2.5. Doxorubicin-Loaded Fe_3_O_4_@CaCO_3_ Synthesis

Briefly, 0.025–3.2 mg Fe_3_O_4_@CaCO_3_ was redispersed in 0.8 mL of deionized water. The 0.1 mL of DOX (100 µg, 1 mg/mL) and 0.1 mL of sodium borate buffer pH 8.0 (10 mM) were added. The mixture was incubated at 25 °C for 12 h under stirring (750 rpm). The nanoparticles were collected using a magnet, washed with sodium borate buffer pH 8.0 (10 mM) three times, and redispersed in sodium borate buffer pH 8.0 (10 mM). The concentration of DOX in the supernatant was determined spectrophotometrically (λ = 480 nm). The amount of the loaded drug was determined as a capacity using the equation: E = (DOX_0_ − DOX)/N. The DOX_0_ and DOX represent the initial and the discard solution amount of DOX (µg), respectively. N denotes the amount of Fe_3_O_4_@CaCO_3_ (mg).

### 2.6. Doxorubicin Release from Fe_3_O_4_@CaCO_3_

The release of DOX was investigated at 25 °C in 1 mL of 100 mM sodium acetate buffer (pH of 4.0 to 6.0) or 10 mM phosphate-buffered saline (pH 7.4) containing Fe_3_O_4_@CaCO_3_ (variable amount of DOX with 0.1 mg of Fe_3_O_4_@CaCO_3_) with constant stirring (750 rpm). The amount of DOX released into the solution was determined using the optical density of the solution at different time points.

### 2.7. Human Serum Albumin and Fe_3_O_4_@CaCO_3_/DOX Interactions

To the solution of human serum albumin (HSA, 0.8–32 mg) in PBS (1.0 mL), Fe_3_O_4_@CaCO_3_/DOX (0.1 mg) was added. The mixture was stirred (750 rpm) at 37 °C for 1, 4, 8, and 24 h. The Fe_3_O_4_@CaCO_3_/DOX nanocomposites coated with HSA were collected using a magnet and redispersed in deionized water. The concentration of the released DOX and HSA were measured spectrophotometrically.

### 2.8. The Cytotoxicity Assay (MTT Test)

Tumor cell lines from human mammary adenocarcinoma MCF-7 and cervical cancer HeLa (Russian Branch of the ETCS, St. Petersburg, Russia) were plated in 96-well culture plates (5 × 10^3^ cells per well) in DMEM medium supplemented with 10% FBS, 1% GlutaMax, and 1% antimycotic antibiotic solution at 37 °C and 5% CO_2_ for 24 h. 

The cytotoxicity studies were performed using a colorimetric assay based on the cleavage of 3-[4,5-dimethylthiazol-2-yl]-2,5-diphenyl-tetrazolium bromide (MTT) by mitochondrial dehydrogenases in viable cells, leading to a blue precipitate of formazan formation [74]. The cells were supplemented with media containing Fe_3_O_4_@CaCO_3_ (0.2–2000 µg/mL), Fe_3_O_4_@CaCO_3_/DOX (0.9–73.4 µg/mL), DOX (0.1–10.0 μM) for 48 h at 37 °C and 5% CO_2_. The cells incubated with the medium were used as a control. After incubation, the medium was removed, and 200 µL of MTT solution (0.25 mg/mL in the culture medium containing 1% of antimycotic antibiotic solution) was added and incubated for 4 h under the same conditions. Afterward, the medium was removed, and formazan was dissolved in 0.1 mL of DMSO. The optical density was measured on a multichannel plate reader Clariostar at 570 nm (peak). The percentage of surviving cells was calculated from the obtained optical density as a percentage of the control values. The half-maximal inhibitory concentration (IC 50) was calculated graphically. All measurements were repeated not less than three times with a standard deviation calculation.

## 3. Results and Discussion

### 3.1. Synthesis and Characterization of Fe_3_O_4_@CaCO_3_

Nanoparticles can accumulate and be retained in tumors from circulating blood due to the enhanced permeability and retention (EPR) effect. However, the optimal size of the nanoparticles should be lower than 150–200 nm to penetrate through vascular structures. On the contrary, nanoparticles smaller than 10–20 nm are rapidly cleared by renal filtration. In this way, this work aimed to synthesize the nanocomposites with optimal size. The synthesis of Fe_3_O_4_@CaCO_3_ nanocomposites was carried out in a two-step procedure by the production of the magnetite core with subsequent carbonate coating (Figure 1). Magnetic iron oxide nanoparticles were synthesized using the classical co-precipitation method. The procedure consists of Fe^2+^/Fe^3+^ salts and surfactant co-precipitation in the presence of the base (see Section 2.3). The widely used surfactant, oleic acid, was chosen for nanoparticle stabilization, allowing high saturation magnetization value [72,75,76,77]. 

For CaCO_3_ layer synthesis, our group’s adapted previously published protocol was utilized [41]. The fabrication of small-sized, porous, and stable CaCO_3_ nanoparticles is a difficult task [78]. Obtaining a stable suspension of calcium carbonate nanoparticles down to 200 nm is still a methodological challenge [41,78]. There are a few synthesis methods that produce magnetic Fe_3_O_4_@CaCO_3_ hybrids with particle size higher than 1 µm [50,51,52,54]. Despite their disadvantages, nanoscale particles are required for drug delivery. Only two research groups presented the synthesis of Fe_3_O_4_@CaCO_3_ nanocomposites as promising multifunctional drug delivery systems [47,49]. In the present work, the mixture of stabilization agents such as polyethylene glycol 2000 (PEG 2000) and Tween 20 in the presence of cell media DMEM was used to reduce the size and increase the stability and monodispersity of CaCO_3_ nanoparticles. The novelty of our approach, apart from using the different compositions of the reaction mixture during the formation phase of the CaCO_3_ layer, is that there is no need to stabilize the composite with additional coatings that alter its characteristics, including sensitivity to pH and drug sorption. PEG-2000 and Tween 20 form a polymeric structure that serves as a template limiting particle growth during nucleation [41]. The amino acids, vitamins, and salts in DMEM presumably stabilize the nanoparticles. PEG-2000 and Tween 20 are well-known amphiphilic surfactants [79,80,81]. Moreover, PEGylation decreases the non-specific interactions with proteins, improves biocompatibility, and prolongs blood circulation time [79]. The obtained nanoscale Fe_3_O_4_ and Fe_3_O_4_@CaCO_3_ were characterized by dynamic light scattering (DLS, Table 1) and transmission electron microscopy (TEM, Figure 2). For comparison, CaCO_3_ nanoparticles were synthesized according to the same procedure [41]. 

To optimize the composition of the composite, the concentration of the magnetic core was varied from 0.45 to 4.5 mg/mL. All samples obtained had sufficient magnetic properties to allow magnetic separation to be applied in all stages of the operation (Figure 3). The surface charge, size, and morphology of nanoparticles are essential factors that influence cell and tissue internalization. Small nanoparticles (5–10 nm) have demonstrated high renal clearance. Large-sized nanoparticles (>200 nm) have a problem passing through cellular membranes [82]. The calculated mean particle diameters by TEM and DLS are summarized in Table 1. According to the DLS, Fe_3_O_4_@CaCO_3_ nanocomposites have a comparable size of 120–130 nm, which is 30 nm higher than the initial Fe_3_O_4_ and optimal for drug delivery applications. However, the diameter of nanoparticles by TEM and DLS highly differ by at least 1.5–2 times, which was previously found for CaCO_3_ nanoparticles [45,49,83]. This phenomena is probably due to the aggregation and swelling occurring in water [83] and the water shell on the nanoparticle surface. Although a significant difference in the size and agglomeration of the nanocomposites can be seen visually on the TEM, the differences in the hydrodynamic diameter by DLS are insignificant. This is probably due to the greater propensity of particles with a high content of magnetite to aggregate during the preparation of samples for TEM [84]. Moreover, as the amount of Fe_3_O_4_ in the sample decreases, the proportion of carbonate component increases, and the degree of particle aggregation decreases (cf. Figure 2C–F), which is clearly expressed in Figure 2F. For further investigation, magnetic nanocomposite with the lowest Fe_3_O_4_ amount (0.45 µg/mL) and the highest carbonate component was chosen (Table 1, Figure 3F and Appendix A). The high similarity between the TEM images of the selected sample (Figure 2F) and the pure CaCO_3_ (Figure 2B) may indicate the absence of a magnetic core. However, the sample has enough magnetic properties, shown using easy magnetic separation in each step of the synthesis (Figure 3). The resulting Fe_3_O_4_@CaCO_3_ nanocomposite has a size by DLS of 121 ± 6 nm (PDI = 0.31 ± 0.01) and ζ-potential (−15.6 ± 0.5 mV).

To confirm the components of oleic acid-modified Fe_3_O_4_@CaCO_3_ nanocomposites, the Fe_3_O_4_ nanoparticles, Fe_3_O_4_@CaCO_3_, and Fe_3_O_4_@CaCO_3_/DOX FTIR spectra were recorded, and the results were presented in Figure 4. The FTIR spectrum of oleic acid-coated Fe_3_O_4_ nanoparticles displayed the characteristic adsorption peaks of oleic acid and Fe_3_O_4_ core at 3421, 2918 (asymmetric -CH_2_ stretch), 2854 (symmetric -CH_2_ stretch), 1626 (-COO^−^ stretch), 1464 (O-H stretch in plane), and 584 cm^−1^ (Fe-O stretch) [55,85]. For Fe_3_O_4_@CaCO_3_, the spectrum shows the same stretches and the appearance of new bands at 1464 (main asymmetric stretch with a shoulder), 1150 (symmetric stretch), 890 (out-of-plane bending), and ~750 cm^−1^ (in-plane bending), corresponding to CO_3_^2−^ [55,86,87]. The nanocomposite stability in an aqueous media is an essential factor for biomedical applications. After the synthesis, Fe_3_O_4_@CaCO_3_ retains stability in deionized water during storage at 7 °C for 5 months without significant changes in the size evaluated by DLS (Appendix A). The further stability of Fe_3_O_4_@CaCO_3_ was analyzed in acetate buffer (pH 5.0, 6.0), PBS buffer (pH 7.4), DMEM, and 10% fetal bovine serum (FBS) using DLS (Appendix A). In acetate buffer (pH 5.0 and 6.0) and FBS, Fe_3_O_4_@CaCO_3_ showed almost the same average size as the initial nanoparticles for at least one week (Appendix A). However, in PBS and DMEM, Fe_3_O_4_@CaCO_3_ immediately resized from 121 nm to 370–390 and 850–890 nm, respectively. No changes occurred during one-week storage (Appendix A). After 8 days of incubation, Fe_3_O_4_@CaCO_3_ nanoparticles were magnetically separated, resuspended in deionized water without sonification, and analyzed using DLS. The hydrodynamic diameter of Fe_3_O_4_@CaCO_3_ was 180 ± 7 nm (PBS) and 106 ± 7 nm (DMEM). Despite the slight changes in size, the material was still less than 200 nm, which is essential for biomedical applications.

### 3.2. Anticancer Drug Doxorubicin (DOX) Loading

The DOX loading efficiency onto the Fe_3_O_4_@CaCO_3_ was studied using UV-vis spectroscopy (480 nm) and fluorescence. The optical density of the buffer solution with a drug was evaluated before and after loading, allowing the calculation of DOX capacity. According to the previously published procedure [41,72], the sodium borate buffer (10 mM, pH 8.0) was chosen for drug loading. The loading was carried out at 25 °C for 12 h. The drug loading on the nanocomposite may be easily seen in the photography as the appearance of red color during the separation procedure on the magnetic rack (Appendix A). The same qualitative data may be obtained by recording the fluorescence and UV-vis spectra of Fe_3_O_4_@CaCO_3_/DOX nanocomposites (Appendix A). Moreover, Fe_3_O_4_@CaCO_3_/DOX can be characterized using FTIR spectroscopy (Figure 4). The same peaks as for Fe_3_O_4_@CaCO_3_ may be observed in the Fe_3_O_4_@CaCO_3_/DOX spectrum (Figure 4). The drug-loading results in intensive specific adsorption peaks of DOX at 2968 (C–H stretch), 1684 (C=O stretch, quinone), 1660 (C=C ring stretch), 1406 (C–C), 1336, 1255, 1221 (=C–O–CH_3_), and 1150 cm^−1^ [88,89].

The capacity was estimated as the amount of DOX bound to 1 mg of nanoparticles. By varying the initial particle concentration (from 0.025 to 3.2 mg/mL), capacity values from 25 to 1900 µg/mg (DOX/Fe_3_O_4_@CaCO_3_) were obtained (Table 2, Figure 5). Figure 5 shows the dependence of Fe_3_O_4_@CaCO_3_ capacity (µg/mg) and DOX loading efficiency (%) on nanoparticle concentration. It can be easily seen on the chart that a high Fe_3_O_4_@CaCO_3_ concentration results in low capacity with a drug high loading efficiency. The graph has a linear relationship in the presented range of concentrations. For excellent drug-loading, a low amount of nanocomposites and the same concentration of DOX are required. However, it is difficult to use such a low concentration of Fe_3_O_4_@CaCO_3_ for Fe_3_O_4_@CaCO_3_/DOX fabrication, which is expressed in the complexity of composites addition, increased procedure volumes, high consumption of the antibiotics, and the complexity of isolation due to the low efficiency of magnetic separation in high solution volume.

The initial Fe_3_O_4_@CaCO_3_ and Fe_3_O_4_@CaCO_3_/DOX samples with a capacity between 25 and 1045 µg/mg have a similar hydrodynamic diameter and ζ-potential (Table 2). The slight decrease in particle size with capacity increase perhaps occurs due to the increase in the packing density of DOX, which may be confirmed by changes in the nanoparticle density measured by TEM (Figure 6). The changes in ζ-potential and particle size in Fe_3_O_4_@CaCO_3_/DOX with a high capacity of 1900 µg/mg are probably associated with a change in the predominant interactions from nanoparticle-DOX type (electrostatic interactions) to DOX-DOX type (hydrophobic interactions) (Appendix A). Furthermore, according to Figure 5, the capacity and efficacy curves intersect at a point of 1045 µg/mg. Such a capacity is considered optimal in terms of drug consumption. However, the difference in intermolecular interactions may lead to changes in drug pH-sensitive release efficiency, which is investigated in Section 3.3.

### 3.3. Doxorubicin Release

The pH-sensitive DOX release efficiency is an essential factor. The release of DOX from various Fe_3_O_4_@CaCO_3_/DOX nanocomposites was investigated at pH 4.0, 5.0, 6.0, and 7.4 (Table 3, Figure 7 and Appendix A). The pH range was selected from plasma pH of 7.4 to acidic, mimicking tumor microenvironment and cell endosomes (pH~5). The concentration of the released DOX was measured spectrophotometrically and using fluorescence with a Clariostar plate reader (BMG Labtech, Ortenberg, Germany). The nanocomposites showed pH-dependent drug distribution. Nanocomposites are presented in Table 3 and Figure 7, with their capacity decrypted at the end of the abbreviation. For example, Fe_3_O_4_@CaCO_3_/DOX25 means DOX-loaded Fe_3_O_4_@CaCO_3_ nanocomposite with 25 µg/mg DOX capacity. We calculated the DOX release efficiency as a percent of the initial DOX amount and in absolute values (µg/mL). As expected, DOX release was more efficient in acidic pH, which is in good correlation with previously published works [41,72]. Figure 7B and Table 3 indicate that the nanocomposites Fe_3_O_4_@CaCO_3_/DOX25-45 are characterized by a 100% release of the drug at pH 5.0. The increase in nanoparticle capacity results in a lower release efficiency of up to 44–70% (Fe_3_O_4_@CaCO_3_/DOX73–295) and 21–23% (Fe_3_O_4_@CaCO_3_/DOX525–1900).

There is a tendency for decrease in release percentage with increasing nanocomposite capacity over the entire pH range studied (Figure 7, Table 3). However, the recalculation to the absolute values (µg/mL, Figure 6A) leads to a clear demonstration of Fe_3_O_4_@CaCO_3_/DOX1900 and Fe_3_O_4_@CaCO_3_/DOX1045 nanocomposites’ achievements. Fe_3_O_4_@CaCO_3_/DOX1900 is 2.7 times more effective than Fe_3_O_4_@CaCO_3_/DOX1045 and 5.4 times more effective than Fe_3_O_4_@CaCO_3_/DOX160 at pH 5.0 (Figure 6A). Moreover, the most efficient nanocomposite at pH 4-5, Fe_3_O_4_@CaCO_3_/DOX1900 showed seven times less DOX release in the physiological pH region, which is 4% of the total amount of the loaded drug. These indicators are promising for further studies of nanoparticles as a container for anticancer drugs. Moreover, at this stage, particles with a capacity of 1900 µg/mL can be distinguished from the obtained nanocomposites as the most effective in terms of the absolute values (µg/mL) of the released drug.

### 3.4. Cellular Toxicity Study of Fe_3_O_4_@CaCO_3_ and Fe_3_O_4_@CaCO_3_/DOX

The cytotoxicity of Fe_3_O_4_@CaCO_3_ and Fe_3_O_4_@CaCO_3_/DOX nanocomposites was studied on HeLa (cervical cancer) and MCF-7 (breast cancer) cell lines. Fe_3_O_4_@CaCO_3_ showed extremely low cytotoxicity in the wide concentration range of up to 2 mg/mL (Figure 8). CaCO_3_ is endowed with high biocompatibility and the absence of hemolytic effect, which was shown by a number of works [39,40,41,42,43,44,90]. Several works show the safety of nanoscale Fe_3_O_4_@CaCO_3_ of up to 0.8 mg/mL [47].

Fe_3_O_4_@CaCO_3_/DOX nanocomposites and DOX cell viability experiments were performed at the same DOX concentration (µM) in cell media or the concentration of nanoparticles (µg/mL) (Figure 9). Fe_3_O_4_@CaCO_3_/DOX25-1900 nanocomposites effectively suppressed cellular activity. The experiment with the same concentration of DOX in the cell media showed comparable cell inhibition for DOX-loaded nanoparticles and free drug (Figure 9, left). However, Fe_3_O_4_@CaCO_3_/DOX1900 indicated higher cell viability, which may be explained by prolonged drug release. Such an effect appears to a lesser degree for other DOX-loaded nanocomposites with a high capacity (Figure 9, left). This fact confirms the data on DOX release efficiency at physiological pH (Table 3, % columns). The diagram in Figure 9 on the right correlates with the release data in absolute values (Table 3, µg/mL columns). These data show the impressive potential of Fe_3_O_4_@CaCO_3_/DOX with a capacity higher than 160 µg/mg. Of course, more pronounced cell inhibition was found for 1045 and 1900 µg/mg capacity.

The half-maximum inhibitory concentration (IC 50) (Table 4) in terms of Fe_3_O_4_@CaCO_3_/DOX concentration (µg/mL) looks consistent and confirms all previously obtained results as well as the persistence of the inhibitory activity of DOX in the nanocomposite. When normalizing the results of the MTT test to the DOX concentration, we observed the best IC 50 in particles with a higher release efficiency in percent (Fe_3_O_4_@CaCO_3_/DOX160), and when normalizing to the concentration of nanoparticles, the best IC 50 in particles with the maximum loading capacity (Fe_3_O_4_@CaCO_3_/DOX1900). The presented work is the first one describing Fe_3_O_4_@CaCO_3_/DOX nanocomposites. When comparing Fe_3_O_4_@CaCO_3_/DOX with nanoscale Fe_3_O_4_/DOX and CaCO_3_/DOX, it can be seen that the developed system is more effective [41,72,91,92,93]. 

### 3.5. Human Serum Albumin Interaction with Fe_3_O_4_@CaCO_3_/DOX

On entering biological fluids, the surface of the nanomaterials is quickly coated with biomolecules forming a bioinspired coating [94]. Serum or cellular proteins can form strongly and weakly adsorbed protein coating, known as “hard and soft nanoparticle corona,” respectively [94]. For instance, the protein corona produces stealth-like properties, improving biodistribution, biocompatibility, cellular interaction, and the recognition of nanoparticles by immune cells [95,96,97,98]. The adsorbed protein may conceal targeting molecules on the nanoparticle surface and dictate the bio-reactivity [98]. In human plasma, a typical nanoparticle corona consists of major proteins such as serum albumin, immunoglobulins, fibrinogen, apolipoproteins, etc. [98,99]. Overall, the most abundant corona protein is serum albumin, which arranges a “crash test” for drugs and nanoparticles [99].

Human serum albumin (HSA) is a major transport human plasma protein. It is important for organism functions and forms covalent and reversible dimers, oligomers, and posttranslational modifications [3,100]. Various albumin-based multifunctional constructions were synthesized for therapy and diagnostic applications [3,101,102,103,104,105]. HSA influences drug pharmacokinetics and pharmacodynamics. Nanomaterials getting into the blood interact with albumin forming a corona on the surface, which highly changes stability, biodistribution, and pharmacokinetics, and reduces the hemolytic effect on red blood cells and toxicity [3,96,98,106,107,108,109]. Furthermore, albumin-coated nanoparticles demonstrated enhanced cell uptake and more efficient tumor targeting due to the EPR effect and interaction with receptors (e.g., gp60, SPARC, etc.) [3]. The next logical step would be the use of albumin corona as a stealth-coating material by fabricating protein-inorganic core nanocomposites in a controlled condition [96]. It is not only critical for “stealthily” properties but also shows the possible material stability or unexpected drug release in blood. The influence of HSA on Fe_3_O_4_@CaCO_3_/DOX nanocomposites’ stability was tested in 10 mM PBS (pH 7.4) at 37 °C for 12 h, modeling the condition in the human body (Table 5). The results indicate DOX release and HSA binding to the nanocomposites. TEM images showed the destruction of the DOX film and the formation of a protein corona on the surface (cf. Figure 6C and Figure 10). Higher physiological albumin concentration (32 mg/mL, 0.48 mM) leads to higher drug release. For the different Fe_3_O_4_@CaCO_3_/DOX, HSA loading efficiency is the same indicating the saturation of the nanocomposites’ surface binding with protein. However, Fe_3_O_4_@CaCO_3_/DOX loses only a small amount of drug, keeping the retained DOX in the nanocomposite (Table 5).

Afterward, the samples were separated and used for the MTT test (Figure 10), which is usually used for primary toxicity studies. However, the MTT assay does not show non-specific interaction with blood and tissue, or chronic toxicity. Compared to Fe_3_O_4_@CaCO_3_/DOX, the albumin-substituted nanocomposites provide much higher cell viability. We assumed that the albumin corona hides DOX molecules, thereby inhibiting acute toxicity. Nevertheless, the therapeutic effect is retained, which may be revealed in cancer tissue (Figure 11). The preformed albumin corona prevents possible non-specific drug release in human plasma, leading to organism-friendly, less toxic nanoplatform. The interaction between nanocomposite and albumin may also reduce systemic toxicity. However, the presented results require further investigation.

## 4. Conclusions

In summary, magnetic Fe_3_O_4_@CaCO_3_ nanocomposites were obtained using the co-precipitation method, yielding a porous superstructure over the magnetic core. TEM and DLS methods demonstrated that the obtained Fe_3_O_4_@CaCO_3_ MNCs were 135 nm in size and possessed narrow size distributions. The resultant MNCs showed high stability in the physiological conditions, slightly acidic pH, cell media, and FBS. The material showed low cytotoxicity and high biocompatibility. We investigated the possible drug doxorubicin (DOX) loading set up using a series of experiments. This method provided a high DOX capacity of up to 1900 µg/mg (DOX/MNPs) with a pH-responsible drug release. The remaining DOX-loaded MNCs were stable in neutral pH and enabled drug extrication in a mild pH medium with high efficiency. The acidic pH mimics the cancer tumor environment and cell organelle media. Due to the suitable size, magnetic properties, and high drug loading capacity, Fe_3_O_4_@CaCO_3_/DOX is suitable for further in vivo experiments. The series of DOX-loaded Fe3O4@CaCO3 MNCs indicated an excellent inhibition of Hela and MCF-7 cell lines, and the IC 50 values were calculated. The cell viability studies of the series of Fe_3_O_4_@CaCO_3_/DOX MNCs with different drug capacities showed the optimal drug loading of 160 µg/mg per DOX concentration and 1900 µg/mg per nanoparticles amount. However, the higher loading of 1900 µg/mg is the perspective for prolonged clinical effects using low-dose MNCs. The IC 50 values (HeLa cells) were calculated as 2.6 µM in terms of DOX and 1.5 µg/mL in terms of the nanoparticles’ amount. The stability experiments for DOX-loaded Fe_3_O_4_@CaCO_3_ in HSA solution indicated the drug release due to the formation of a protein corona. An experiment should be conducted for any composites that will interact with HSA and other plasma proteins. We assumed that HSA pushed out the DOX molecules, interacting with the MNCs’ surface with a certain capacity. Afterward, the sorption was disrupted, yielding a stable construction, which will not release DOX easily in plasma media. Moreover, the new layer of albumin super coating may increase tumor capture of drug-loaded MNCs. HSA coating increases the colloidal stability, prolongs blood circulation time, and prevents non-specific adsorption of blood components. HSA increases the efficiency of tissue and cell targeting due to the EPR effect and receptor interaction. Finally, DOX-loaded MNCs may offer a high potential for pH-sensitive nanotheranostic areas for drug-resistance cancer treatment. 

To sum up, the proposed excellent, biocompatible Fe_3_O_4_@CaCO_3_ nanocomposite is a good platform for drug delivery and disease treatment. Despite the promising results in the synthesis, drug-loading, and release of Fe_3_O_4_@CaCO_3_, numerous challenges must be addressed for cancer treatment. The optimization of the structure concerning magnetic resonance imaging or hyperthermia properties targeted magnetic transport, and address molecule modification should be carried out. Extensive in vitro or cell experiments should be conducted, involving toxicity, targeting efficacy, and biocompatibility. After these important stages, the functional nanoplatform may be applied to in vivo studies for clinical prospects evaluation. Meanwhile, the Fe_3_O_4_@CaCO_3_ nanoplatform may be considered for various applications of MNCs [35]. 

## Figures and Tables

**Figure 1 pharmaceutics-15-00771-f001:**
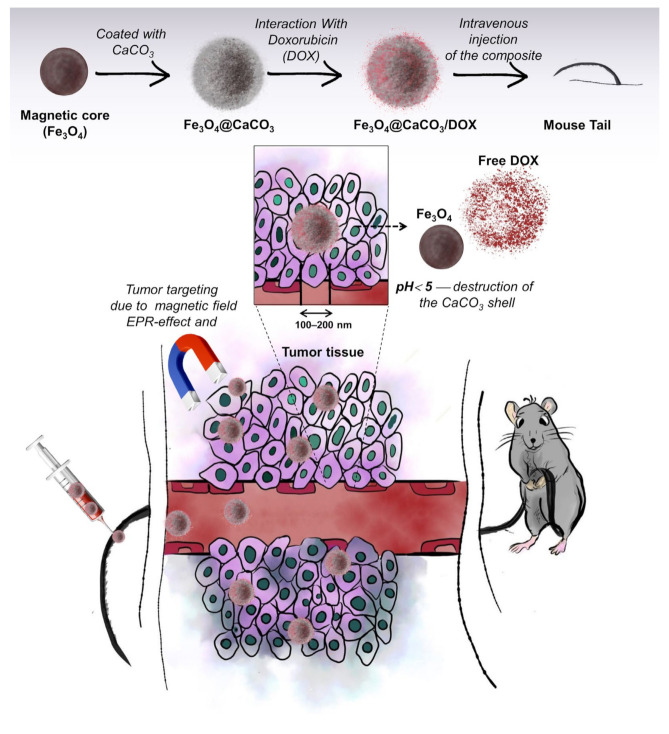
Synthesis of Fe_3_O_4_@CaCO_3_ and doxorubicin-loaded (Fe_3_O_4_@CaCO_3_/DOX) nanocomposites and their distribution through the bloodstream to the tumors.

**Figure 2 pharmaceutics-15-00771-f002:**
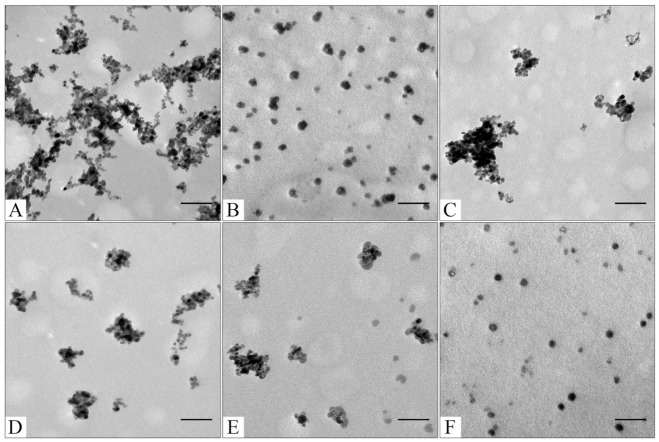
TEM images of sourced Fe_3_O_4_ (**A**); CaCO_3_ (**B**); Fe_3_O_4_@CaCO_3_ obtained using 4.5 mg/mL (**C**); 1.8 mg/mL (**D**); 0.9 mg/mL (**E**); 0.45 mg/mL of Fe_3_O_4_ (**F**). The bar indicates 100 nm.

**Figure 3 pharmaceutics-15-00771-f003:**
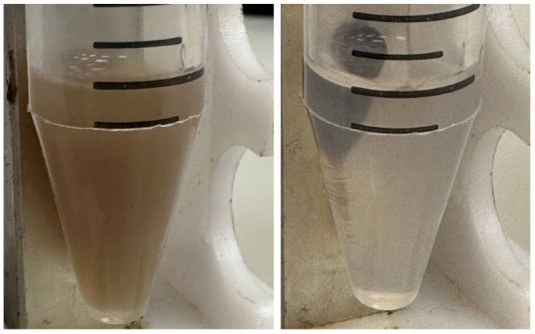
Magnetic separation of Fe_3_O_4_@CaCO_3_ nanocomposite (0.45 mg/mL of Fe_3_O_4_ synthesis) immediately (**left**) and after 15 s (**right**).

**Figure 4 pharmaceutics-15-00771-f004:**
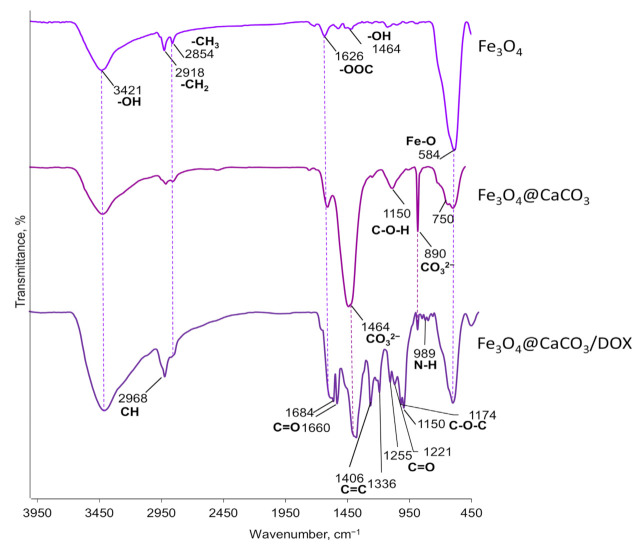
FTIR spectra of Fe_3_O_4_, Fe_3_O_4_@CaCO_3,_ and Fe_3_O_4_@CaCO_3_/DOX nanocomposites.

**Figure 5 pharmaceutics-15-00771-f005:**
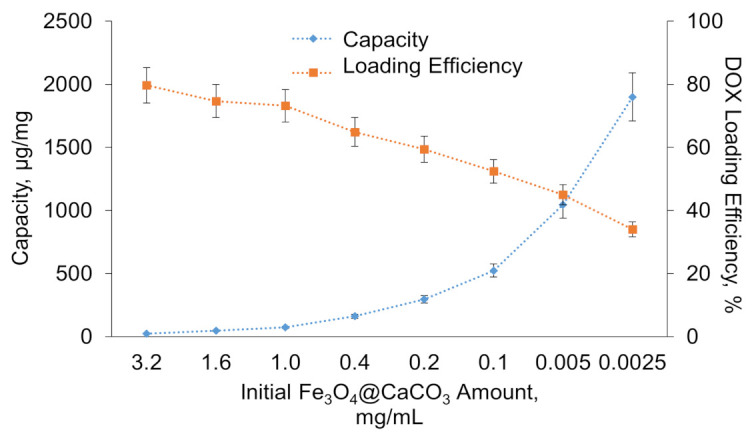
Dependence of Fe_3_O_4_@CaCO_3_ capacity (µg/mg) and DOX loading efficiency (%) on nanoparticle concentration.

**Figure 6 pharmaceutics-15-00771-f006:**
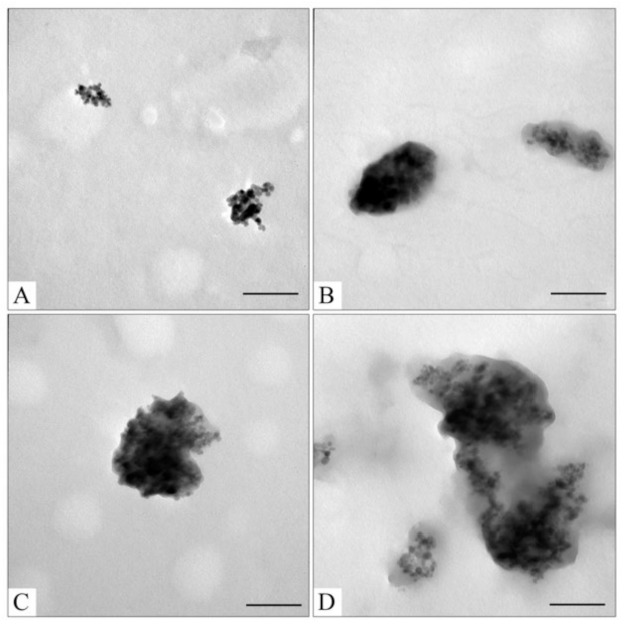
TEM images of Fe_3_O_4_@CaCO_3_/DOX73 (**A**); Fe_3_O_4_@CaCO_3_/DOX295 (**B**); Fe_3_O_4_@CaCO_3_/DOX525 (**C**); Fe_3_O_4_@CaCO_3_/DOX1045 (**D**). The bar indicates 100 nm.

**Figure 7 pharmaceutics-15-00771-f007:**
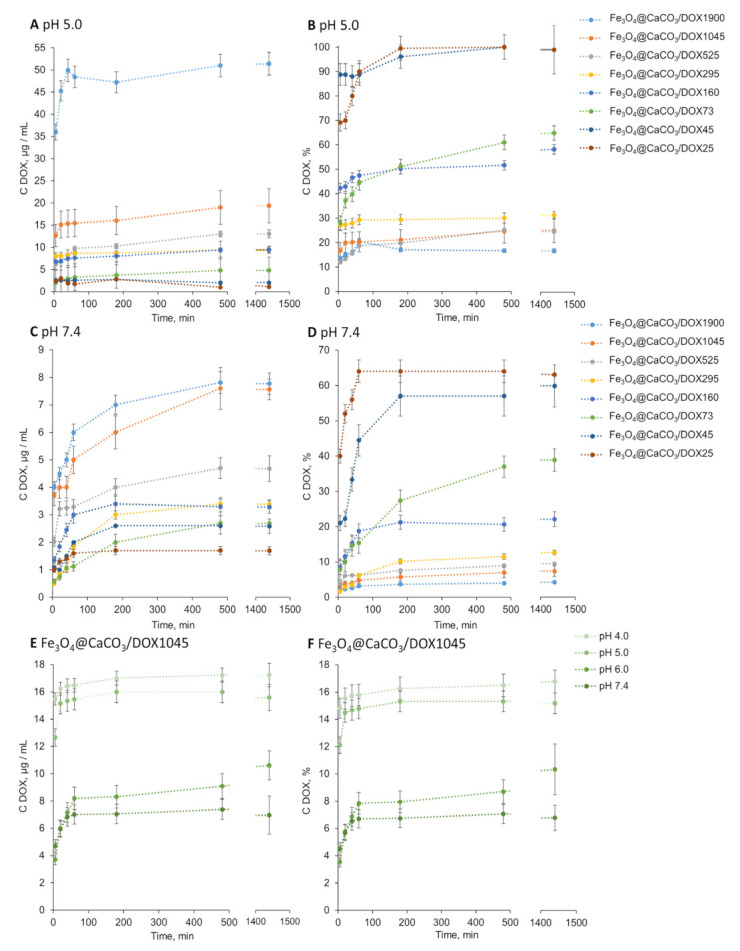
DOX release from Fe_3_O_4_@CaCO_3_/DOX with capacity from 25 to 1900 µg/mg at pH 5.0 (**A**,**B**) and pH 7.4 (**C**,**D**) at 25 °C. DOX release from Fe_3_O_4_@CaCO_3_/DOX with capacity 1045 µg/mg at pH 4.0–7.4 at 25 °C (**E**,**F**).

**Figure 8 pharmaceutics-15-00771-f008:**
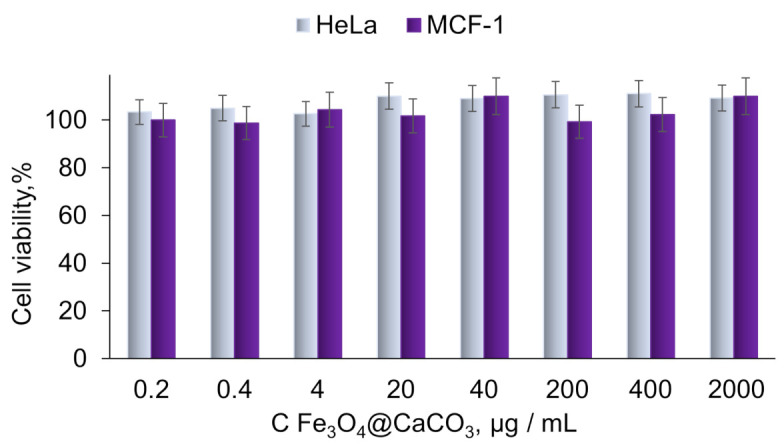
Cell viability assay using MTT. HeLa and MCF-7 cells were incubated with Fe_3_O_4_@CaCO_3_ for 48 h.

**Figure 9 pharmaceutics-15-00771-f009:**
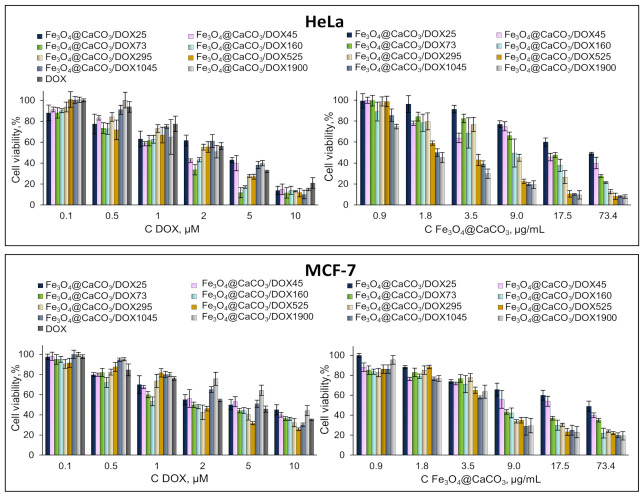
Cell viability assay using MTT. HeLa and MCF-7 cells were incubated with Fe_3_O_4_@CaCO_3_/DOX and free DOX for 48 h.

**Figure 10 pharmaceutics-15-00771-f010:**
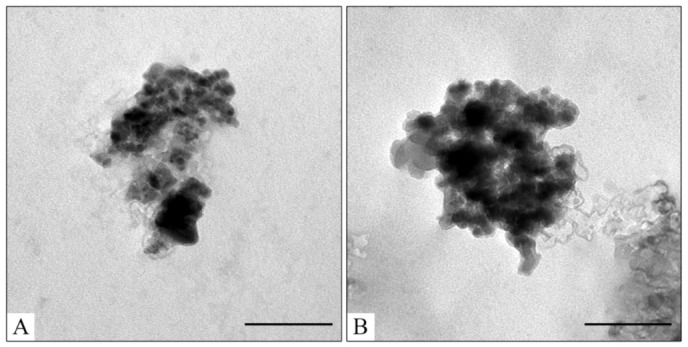
TEM images of Fe_3_O_4_@CaCO_3_/DOX525-HSA3.2 (**A**) and Fe_3_O_4_@CaCO_3_/DOX525-HSA32 (**B**). The bar indicates 100 nm.

**Figure 11 pharmaceutics-15-00771-f011:**
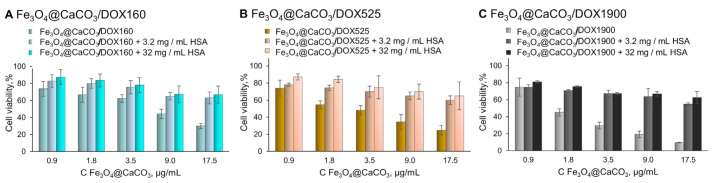
Cell viability assay using MTT. MCF-7 cells were incubated with Fe_3_O_4_@CaCO_3_/DOX and Fe_3_O_4_@CaCO_3_/DOX-HSA for 48 h.

**Table 1 pharmaceutics-15-00771-t001:** Diameter by DLS and TEM, PDI, and ζ-potential of Fe_3_O_4_ and CaCO_3_ nanoparticles and different Fe_3_O_4_@CaCO_3_ nanocomposites using a variable amount of magnetic core from 0.45 to 4.5 mg/mL for synthesis.

Initial Amount Fe_3_O_4_, µg/mL	Hydrodynamic Diameter by DLS, nm	Diameter by TEM, nm		PDI	ζ-Potential, mV	TEM Image (Figure 2)
		NP	Agglomerate			
4.5	130 ± 2	13 ± 2	107 ± 41	0.211 ± 0.006	−15.0 ± 0.3	C
1.8	118 ± 3	16 ± 2	68 ± 20	0.26 ± 0.01	−14.9 ± 0.3	D
0.9	130 ± 9	18 ± 3	61 ± 19	0.248 ± 0.004	−15.5 ± 0.4	E
0.45	121 ± 6	22 ± 1	—	0.31 ± 0.01	−15.6 ± 0.5	F
Initial Fe_3_O_4_	88 ± 5	11 ± 4	91 ± 23	0.221 ± 0.004	21.6 ± 0.7	A
Control CaCO_3_	204 ± 8	26 ± 3	—	0.14 ± 0.02	−17.3 ± 0.4	B

**Table 2 pharmaceutics-15-00771-t002:** Diameter by DLS and TEM, PDI, ζ-potential, and capacity of DOX-loaded nanocomposites Fe_3_O_4_@CaCO_3_/DOX.

Fe_3_O_4_@CaCO_3_ Amount, mg/mL	Hydrodynamic Diameter by DLS, nm	Diameter by TEM, nm			PDI	ζ-Potential, mV	Capacity, µg/mg	DOX Loading Efficiency, %	TEM Image Figure 6
		NP	Agglomerate	Shell					
3.2	133 ± 3	—			0.234 ± 0.005	−14.7 ± 0.7	25± 1	79.7 ± 0.2	—
1.6	130 ± 2	—			0.192 ± 0.008	−13.0 ± 0.2	45 ± 2	74.7 ± 0.1	—
1.0	128 ± 3	10 ± 2	75 ± 13	4.6 ± 0.3	0.144 ± 0.006	−18.8 ± 0.6	73.2 ± 0.4	73.2 ± 0.4	A
0.40	129 ± 3	—			0.130 ± 0.002	−14.9 ± 0.4	160 + 2	64.9 ± 0.6	—
0.20	135 ± 5	11 ± 3	125 ± 31	18 ± 2	0.14 ± 0.01	−15.0 ± 0.3	295 + 2	59.4 ± 0.3	B
0.10	111 ± 3	9 ± 3	149 ± 45	46 ± 9	0.30 ± 0.01	−16.5 ± 0.3	525 ± 3	52.4 ± 0.4	C
0.050	113 ± 5	9 ± 5	201 ± 35	158 ± 46	0.28 ± 0.03	−12.4 ± 0.2	1045 ± 10	45 ± 2	D
0.025	105 ± 3	—			0.29 ± 0.01	−19.0 ± 0.5	1900 ± 27	34 ± 4	—
Initial Fe_3_O_4_@CaCO_3_	121 ± 6	—			0.31 ± 0.01	−15.6 ± 0.5	—	—	—

**Table 3 pharmaceutics-15-00771-t003:** DOX release efficiency from Fe_3_O_4_@CaCO_3_/DOX at pH 4.0–7.4 at 25 °C for 24 h.

Sample Abbreviation with Capacity	pH 4.0		pH 5.0		pH 6.0		pH 7.4	
	C DOX, %	C DOX, µg/mL	C DOX, %	C DOX, µg/mL	C DOX, %	C DOX, µg/mL	C DOX, %	C DOX, µg/mL
Fe_3_O_4_@CaCO_3_/DOX25	98 ± 4	2.4 ± 0.1	99 ± 1	2.48 ± 0.03	68 ± 3	1.69 ± 0.05	60 ± 4	1.5 ± 0.1
Fe_3_O_4_@CaCO_3_/DOX45	94 ± 6	4.2 ± 0.3	98 ± 2	4.41 ± 0.09	49 ± 4	2.2 ± 0.2	57 ± 4	2.6 ± 0.1
Fe_3_O_4_@CaCO_3_/DOX73	71 ± 5	5.2 ± 0.4	65 ± 4	4.8 ± 0.3	38 ± 1	2.8 ± 0.1	37 ± 4	2.7 ± 0.1
Fe_3_O_4_@CaCO_3_/DOX160	65 ± 6	10 ± 1	59 ± 5	9.4 ± 0.8	26 ± 2	4.2 ± 0.3	21 ± 2	3.4 ± 0.1
Fe_3_O_4_@CaCO_3_DOX295	44 ± 4	13 ± 1	32 ± 3	9.4 ± 0.9	17 ± 1	5.0 ± 0.3	12 ± 2	3.5 ± 0.1
Fe_3_O_4_@CaCO_3_/DOX525	23 ± 2	12 ± 1	25 ± 2	13 ± 1	11.4 ± 0.4	6.0 ± 0.2	9 ± 3	4.7 ± 0.2
Fe_3_O_4_@CaCO_3_/DOX1045	23 ± 2	24 ± 2	25 ± 3	26 ± 3	11.1 ± 0.8	12.1 ± 0.8	7 ± 2	7.3 ± 0.2
Fe_3_O_4_@CaCO_3_/DOX1900	21 ± 2	44 ± 4	17 ± 1	32 ± 2	6.8 ± 0.5	13 ± 1	4 ± 1	7.6 ± 0.3

**Table 4 pharmaceutics-15-00771-t004:** IC 50 of different types of Fe_3_O_4_@CaCO_3_/DOX and free DOX on HeLa and MCF-7 cells.

Sample	HeLa		MCF-7	
	IC 50, µM DOX	IC 50, µg/mL Fe_3_O_4_@CaCO_3_/DOX	IC 50, µM DOX	IC 50, µg/mL Fe_3_O_4_@CaCO_3_/DOX
Fe_3_O_4_@CaCO_3_/DOX25	3.0	49.5	5.2	60.0
Fe_3_O_4_@CaCO_3_/DOX45	1.8	22.0	4.5	41.6
Fe_3_O_4_@CaCO_3_/DOX73	1.6	13.8	2.5	16.4
Fe_3_O_4_@CaCO_3_/DOX160	**1.2**	7.0	**2.0**	9.9
Fe_3_O_4_@CaCO_3_/DOX295	2.3	5.8	2.2	7.6
Fe_3_O_4_@CaCO_3_/DOX525	2.4	2.4	3.1	4.3
Fe_3_O_4_@CaCO_3_/DOX1045	2.5	1.9	5.2	4.1
Fe_3_O_4_@CaCO_3_/DOX1900	2.6	**1.5**	5.7	**3.5**
DOX	2.8	—	3.1	—

**Table 5 pharmaceutics-15-00771-t005:** HSA loading on Fe_3_O_4_@CaCO_3_/DOX nanocomposites.

Sample	Initial C HSA mg/mL	Remaining C DOX, %	DOX capacity in Fe_3_O_4_@CaCO_3_ after Incubation, µg/mg	HSA Loading Efficiency, %
Fe_3_O_4_@CaCO_3_/DOX160	3.2	20 ± 2	32 ± 3	16 ± 1
	32	6 ± 1	10 ± 2	18 ± 2
Fe_3_O_4_@CaCO_3_/DOX525	3.2	65 ± 2	341 ± 10	13 ± 1
	32	60 ± 3	315 ± 16	16 ± 1
Fe_3_O_4_@CaCO_3_/DOX1900	3.2	71 ± 2	1539 ± 45	12.3 ± 0.7
	32	64 ± 1	1215 ± 19	18.5 ± 0.5

## Data Availability

The data are contained in the article.

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
