# Peer review of "pH-Responsible Doxorubicin-Loaded Fe3O4@CaCO3 Nanocomposites for Cancer Treatment"

_pharmaceutics, 2023, doi:10.3390/pharmaceutics15030771_

Round 1

Reviewer 1 Report

1. “we reported a novel reproducible synthesis of core-shell inorganic …” I think is method for syntheses MNCs is not novel…

2. From the TEM, I cannot check this material is core-shell.

3.“MNCs have widespread applications in drug delivery, magnetic field influenced transport, magnetic resonance imaging, hyperthermia therapy, theranostics, magnetic separation, and biosensor areas. Some updated and current refs could be considered, such as Colloid Surface B, 2022, 213, 112432 and Dalton Trans., 2021, 50, 18016–18026

4. Please also provide the BET before and after loading DOX.

5. Please also do the IR test before and after loading DOX. Some refs could be cited, such as Micropor. Mesopor. Mat, 341(2022) 112098. Moreover, at this stage, particles with a capacity of 1900 301 μg/mL can be distinguished from the obtained nanocomposites as the most effective in 302 terms of the absolute values (μg/mL) of the released drug. The ref could be cited, such as Inorganics, 10(2022) 202.

6. Give some examples and compare the similar wok on this Cellular Toxicity Study

Author Response

Thank you for the valuable suggestions and comments. We have carefully examined the comments and suggestions and revised the manuscript accordingly. We presented the word file with track changes. Please find as follows the responses to the comments. Please note that all the comments are bold-faced, and the authors' reply follows immediately below the comments.

  1. “we reported a novel reproducible synthesis of core-shell inorganic …” I think is method for syntheses MNCs is not novel…

Thank you for your comments! We have added the information to the 3.1 subsection. There are a few magnetic Fe3O4@CaCO3 hybrids with particles size higher than 1 µm synthesis methods [1–4]. Despite their advantages, nanoscale particles are required for drug delivery. Only two research groups presented the synthesis of Fe3O4@CaCO3 nanocomposites as promising multifunctional drug delivery systems [5,6]. However, our synthesis procedure is not copied from previous ones. We have used the previously published synthesis procedure for Fe3O4. The CaCO3 layer was adapted from Popova V. et al. [7], which is a previously published by our group novel synthesis procedure for nanoscale CaCO3. The novelty of our approach, apart from using the different composition of the reaction mixture during the formation phase of the calcium carbonate layer, is that there is no need to stabilize the composite with additional coatings that alter its characteristics, including sensitivity to pH.

  1. Serov, N.; Prilepskii, A.; Sokolov, A.; Vinogradov, V. Synthesis of Plasmin-Loaded Fe3O4@CaCO3 Nanoparticles: Towards Next-Generation Thrombolytic Drugs. ChemNanoMat 2019, 5, 1267–1271, doi:10.1002/cnma.201900359.
  2. Xue, J.; Li, X.; Li, Q.; Lyu, J.; Wang, W.; Zhuang, L.; Xu, Y. Magnetic Drug-Loaded Osteoinductive Fe3O4/CaCO3 Hybrid Microspheres System: Efficient for Sustained Release of Antibiotics. J. Phys. D. Appl. Phys. 2020, 53, doi:10.1088/1361-6463/ab7bb2.
  3. Li, F.H.; Tang, N.; Wang, Y.Q.; Zhang, L.; Du, W.; Xiang, J.; Cheng, P.G. Synthesis and Characterization of Magnetic Carriers Based on Immobilized Enzyme. IOP Conf. Ser. Mater. Sci. Eng. 2018, 359, doi:10.1088/1757-899X/359/1/012044.
  4. Han, P.; Jiang, Z.; Wang, X.; Wang, X.; Zhang, S.; Shi, J.; Wu, H. Facile Preparation of Porous Magnetic Polydopamine Microspheres through an Inverse Replication Strategy for Efficient Enzyme Immobilization. J. Mater. Chem. B 2015, 3, 7194–7202, doi:10.1039/c5tb01094b.
  5. Vavaev, E.S.; Novoselova, M.; Shchelkunov, N.M.; German, S.; Komlev, A.S.; Mokrousov, M.D.; Zelepukin, I. V.; Burov, A.M.; Khlebtsov, B.N.; Lyubin, E. V.; et al. CaCO3Nanoparticles Coated with Alternating Layers of Poly-L-Arginine Hydrochloride and Fe3O4Nanoparticles as Navigable Drug Carriers and Hyperthermia Agents. ACS Appl. Nano Mater. 2022, 5, 2994–3006, doi:10.1021/acsanm.2c00338.
  6. Wang, P.; Xue, J.; Wu, S.; Pei, Y.; Xu, L.; Wang, Y. Cell-Friendly Isolation and PH-Sensitive Controllable Release of Circulating Tumor Cells by Fe3O4@CaCO3 Nanoplatform. Adv. Mater. Interfaces 2021, 8, 1–11, doi:10.1002/admi.202101191.
  7. Popova, V.; Poletaeva, Y.; Pyshnaya, I.; Pyshnyi, D.; Dmitrienko, E. Designing PH-Dependent Systems Based on Nanoscale Calcium Carbonate for the Delivery of an Antitumor Drug. Nanomaterials 2021, 11, 2794.
  8. From the TEM, I cannot check this material is core-shell.

Thank you for your valuable comments. We have changed the text though the paper.

3.“MNCs have widespread applications in drug delivery, magnetic field influenced transport, magnetic resonance imaging, hyperthermia therapy, theranostics, magnetic separation, and biosensor areas. Some updated and current refs could be considered, such as Colloid Surface B, 2022, 213, 112432 and Dalton Trans., 2021, 50, 18016–18026

Thank you for the suggestion. We have added the references.

  1. Please also provide the BET before and after loading DOX.

Thank you for your valuable comments. However, we haven’t measured such characteristics. For our work, an important factor is drug loading efficiency. Doxorubicin has enough large chemical structure, which not provide clearly correlation with gas physical adsorption, BET experiments. In this way, we gone forward with capacity, loading efficiency, and release studies.

  1. Please also do the IR test before and after loading DOX. Some refs could be cited, such as Micropor. Mesopor. Mat, 341(2022) 112098. Moreover, at this stage, particles with a capacity of 1900 301 μg/mL can be distinguished from the obtained nanocomposites as the most effective in 302 terms of the absolute values (μg/mL) of the released drug. The ref could be cited, such as Inorganics, 10(2022) 202.

Thank you for your valuable comments. We have inserted FTIR data (Figure 4) with a discussion. However, the references of MOFs are beyond this research.

  1. Give some examples and compare the similar wok on this Cellular Toxicity Study

Thank you for your valuable comments. We have revised the discussion in subsection 3.4 and have added some references.

Reviewer 2 Report

The research article entitled “pH-responsible Doxorubicin-loaded Fe3O4@CaCO3 Nanocomposites for Cancer Treatment” focused on the preparations of Doxorubicin-loaded Fe3O4@CaCO3 Nanocomposites and evaluate the simulated pH-responsible of DOX for targeted and controlled delivery for Cancer Treatment, Hemocompatibility and in-vitro biocompatibility need to provided, etc. There are many grammatical and sentence errors in the article, and the language organization needs to be improved. For these reasons, I conclude that the paper is not suitable for its current form and is recommended for Major publication.

1.      Authors need to perform XRD and FTIR to confirm the formation of Fe3O4@CaCO3 Nanocomposites.  

2.      Authors also need to provide HR-TEM (5 nm Scale), EDX as well as EDS mapping the shell-core spatial arrangement of Fe3O4@CaCO3 Nanocomposites. Thus, size and shape can be confirmed.

3.      The loading of DOX on Fe3O4@CaCO3 needs to be confirmed using FTIR analysis.

4.      Hemocompatibility and in-vitro biocompatibility of Fe3O4@CaCO3 need to be studied.

5.      Figure 6 needs to be represented as a line graph for a better understanding of readers.

6.      Typographic errors need to be corrected. The language and grammar used throughout the manuscript need to be improved

7.      The introduction needs to be improved on why Fe3O4 is important in cancer treatment, and how it helps in targeting, etc. refer https://doi.org/10.1002/adfm.201301659

https://doi.org/10.1016/j.ijpharm.2015.03.070

Also, citations need to be provided important quotes such as Line 66-67: The DOX-loaded MNCs should be stable at a plasma pH of ~ 7.4 and have efficient drug release in cancer tissue (pH of 5–5.5). refer https://doi.org/10.1073/pnas.1003919107

https://doi.org/10.3390/polym14194128

https://doi.org/10.1186%2F1475-2867-13-89

https://doi.org/10.1016/j.biomaterials.2016.09.025

Author Response

Thank you for the valuable suggestions and comments. We have carefully examined the comments and suggestions and revised the manuscript accordingly. We presented the word file with track changes. Please find as follows the responses to the comments. Please note that all the comments are bold-faced, and the authors' reply follows immediately below the comments.

  1. Authors need to perform XRD and FTIR to confirm the formation of Fe3O4@CaCO3 Nanocomposites. 

Thank you for your valuable comments. We have inserted FTIR data (Figure 4) with a discussion.

  1. Authors also need to provide HR-TEM (5 nm Scale), EDX as well as EDS mapping the shell-core spatial arrangement of Fe3O4@CaCO3 Nanocomposites. Thus, size and shape can be confirmed.

Thank you for your valuable comments. However, we haven’t got opportunity to use high-resolution TEM. We have one more method as dynamic light scattering (DLS), which clearly show the nanocomposite size.

We have used the TEM studies as one more method to confirm the nanoparticle's behavior. We have deleted the core-shell statement.

  1. The loading of DOX on Fe3O4@CaCO3 needs to be confirmed using FTIR analysis.

Thank you for your valuable comments. We have inserted FTIR data (Figure 4) with a discussion and new UV-vis. spectra of Fe3O4@CaCO3/DOX and its color photography in Supplementary materials.

  1. Hemocompatibility and in-vitro biocompatibility of Fe3O4@CaCO3 need to be studied.

Hemocompatibility is an extensive biocompatibility study. We haven’t got an opportunity to present it in this work without ethics committee approval and further studies. However, calcium carbonate coating is highly biocompatible. It hasn’t got an expected pronounced hemolytic effect.

Lin, J.; Huang, L.; Xiang, R.; Ou, H.; Li, X.; Chen, A.; Liu, Z. Blood Compatibility Evaluations of CaCO3 Particles. Biomed. Mater. 2021, 16, 055010.

  1. Figure 6 needs to be represented as a line graph for a better understanding of readers.

Thank you for the suggestion. We have changed Figure 6 (now Figure 7).

  1. Typographic errors need to be corrected. The language and grammar used throughout the manuscript need to be improved.

Thank you for the suggestion. We have tried to improve the text.

  1. The introduction needs to be improved on why Fe3O4 is important in cancer treatment, and how it helps in targeting, etc. refer https://doi.org/10.1002/adfm.201301659

https://doi.org/10.1016/j.ijpharm.2015.03.070

Also, citations need to be provided important quotes such as Line 66-67: The DOX-loaded MNCs should be stable at a plasma pH of ~ 7.4 and have efficient drug release in cancer tissue (pH of 5–5.5). refer https://doi.org/10.1073/pnas.1003919107

https://doi.org/10.3390/polym14194128

https://doi.org/10.1186%2F1475-2867-13-89

https://doi.org/10.1016/j.biomaterials.2016.09.025

Thank you for the suggestion. We have inserted some references.

Reviewer 3 Report

The article entitled pH-responsible Doxorubicin-loaded Fe3O4@CaCO3 Nanocomposites for Cancer Treatment is a document of interesting subject matter due the interest in new nano-carriers for drug delivery.  This manuscript presents new useful information, the results are well documented, and the experimental technique and processing of the data meet high standards.

Therefore, I think this paper is a fine contribution to the journal. However, it needs some major changes before being accepted. Make the following corrections:

1.      Please modify the paper title by inputting “cancer name” to being more attractive. I think current title doesn’t reflect the paper contents.

2.      Please Add 1 or 2 lines as per novelty of work for indicating impact the pH-responsible Doxorubicin-loaded Fe3O4@CaCO3 Nanocomposites for Cancer Treatment in the 'Abstract' section.

3.      'Introduction' section not enough discussed with recent updates on  nanocarriers for drug delivery to treat cancers  should follow the cited links given:

-  https://doi.org/10.3390/polym14122403

- https://doi.org/10.1007/s00289-020-03354-6

- https://doi.org/10.3390/molecules27175369

- https://doi.org/10.2174/2211738510666220210105113

- https://doi.org/10.1007/s00289-023-04688-7

- https://doi.org/10.3390/antiox12020237

4.      Please input reference for “2.5. Doxorubicin-loaded Fe3O4@CaCO3 Synthesis”.

5.      Please try to increase resolution of Figure 6.

6.      Line 13, “Herein, the core–shell Fe3O4@CaCO3 nanocomposites were successfully obtained by coprecipitation using a magnetic core nanoscale oleic acid-modified Fe3O4, PEG-2000, and Tween 20 as a template for porous CaCO3 coating.” Please re-write this sentence to more clarify.

7.      Line 218, “This is probably due to the greater propensity of particles with a high content of magnetite to aggregate during the preparation of samples for TEM.”, please input reference for this issue.

8.      Also, in other sections of results, there is a lack of thorough discussion on the results. The authors should be more informative and provide more comparison between the results of the current work with former studies.

9.      Why did the authors choose the PEG-2000, and Tween 20 in the paper as a template for porous CaCO3 coating?

10.  Table 1 and Table 2, please input size by the TEM images in the Table 1 and Table 2 to compare with size reported by the DLS.

11.  Please try to more discussion on relationship between size and release  studies of drug- loaded nanocomposites.

12.  Does the size and structure of the NP play an important role?It is noteworthy that 100 nm particles or larger generally do not penetrate well throughout the tumour mass, and smaller nanoparticles do not accumulate sufficiently in the tumour vasculature by the enhanced permeability and retention (EPR) effect and do not achieve good tumour penetration.

13.  Line 336, discussion regarding” 3.5. Human Serum Albumin Interaction with Fe3O4@CaCO3/DOX” is very short. Please do more discussion with supportive literature review.

14.  How likely is the new agent approved by FDA?

  1.  It is better to present the data of Fe3O4, CaCO3 in treatment.

16.  In the conclusion, provide a brief explanation about the future perspective of the developed carrier and how it can be modified to exhibit better performance. In the other words, the authors should do the analysis the conclusion section must clearly establish a strong correlation with the proposed topic.

17.  Your abstract should clearly state the essence of the problem you are addressing, what you did and what you found and recommend. That will help a prospective reader of the abstract to decide if they wish to read the entire article.

18.  Why the authors did not fitted the release data in known release models? 

19.  The encapsulation efficiency discussion is very short.

As it stands, concern author should give another chance to revise their article and should highlight in the revised manuscript text, so far recommended that the article is mandatory for 'Major revision'.

Author Response

Thank you for the valuable suggestions and comments. We have carefully examined the comments and suggestions and revised the manuscript accordingly. We presented the word file with track changes. Please find as follows the responses to the comments. Please note that all the comments are bold-faced, and the authors' reply follows immediately below the comments.

  1. Please modify the paper title by inputting “cancer name” to being more attractive. I think current title doesn’t reflect the paper contents.

Thank you for your valuable comments. However, such nanocomposites may be used for various cancer treatments. We have shown the potential therapeutic effect on two cell lines.

  1. Please Add 1 or 2 lines as per novelty of work for indicating impact the pH-responsible Doxorubicin-loaded Fe3O4@CaCO3 Nanocomposites for Cancer Treatment in the 'Abstract' section.

Thank you for your valuable comments. We have revised the text.

  1. 'Introduction' section not enough discussed with recent updates on nanocarriers for drug delivery to treat cancers  should follow the cited links given:

-  https://doi.org/10.3390/polym14122403

- https://doi.org/10.1007/s00289-020-03354-6

https://doi.org/10.3390/molecules27175369

https://doi.org/10.2174/2211738510666220210105113

https://doi.org/10.1007/s00289-023-04688-7

- https://doi.org/10.3390/antiox12020237

Thank you for your valuable comments. We have added the references.

  1. Please input reference for “2.5. Doxorubicin-loaded Fe3O4@CaCO3 Synthesis”.

Thank you for the suggestion. However, the synthesis of Fe3O4@CaCO3 by a new method. We have added the information to the 3.1 subsection. There are a few magnetic Fe3O4@CaCO3 hybrids with particles size higher than 1 µm synthesis methods [1–4]. Despite their advantages, nanoscale particles are required for drug delivery. Only two research groups presented the synthesis of Fe3O4@CaCO3 nanocomposites as promising multifunctional drug delivery systems [5,6]. However, our synthesis procedure is not copied from previous ones. We have used the previously published synthesis procedure for Fe3O4. The CaCO3 layer was adapted from Popova V. et al. [7], which is a previously published by our group novel synthesis procedure for nanoscale CaCO3. The novelty of our approach, apart from using the different composition of the reaction mixture during the formation phase of the calcium carbonate layer, is that there is no need to stabilize the composite with additional coatings that alter its characteristics, including sensitivity to pH.

  1. Serov, N.; Prilepskii, A.; Sokolov, A.; Vinogradov, V. Synthesis of Plasmin-Loaded Fe3O4@CaCO3 Nanoparticles: Towards Next-Generation Thrombolytic Drugs. ChemNanoMat 2019, 5, 1267–1271, doi:10.1002/cnma.201900359.
  2. Xue, J.; Li, X.; Li, Q.; Lyu, J.; Wang, W.; Zhuang, L.; Xu, Y. Magnetic Drug-Loaded Osteoinductive Fe3O4/CaCO3 Hybrid Microspheres System: Efficient for Sustained Release of Antibiotics. J. Phys. D. Appl. Phys. 2020, 53, doi:10.1088/1361-6463/ab7bb2.
  3. Li, F.H.; Tang, N.; Wang, Y.Q.; Zhang, L.; Du, W.; Xiang, J.; Cheng, P.G. Synthesis and Characterization of Magnetic Carriers Based on Immobilized Enzyme. IOP Conf. Ser. Mater. Sci. Eng. 2018, 359, doi:10.1088/1757-899X/359/1/012044.
  4. Han, P.; Jiang, Z.; Wang, X.; Wang, X.; Zhang, S.; Shi, J.; Wu, H. Facile Preparation of Porous Magnetic Polydopamine Microspheres through an Inverse Replication Strategy for Efficient Enzyme Immobilization. J. Mater. Chem. B 2015, 3, 7194–7202, doi:10.1039/c5tb01094b.
  5. Vavaev, E.S.; Novoselova, M.; Shchelkunov, N.M.; German, S.; Komlev, A.S.; Mokrousov, M.D.; Zelepukin, I. V.; Burov, A.M.; Khlebtsov, B.N.; Lyubin, E. V.; et al. CaCO3Nanoparticles Coated with Alternating Layers of Poly-L-Arginine Hydrochloride and Fe3O4Nanoparticles as Navigable Drug Carriers and Hyperthermia Agents. ACS Appl. Nano Mater. 2022, 5, 2994–3006, doi:10.1021/acsanm.2c00338.
  6. Wang, P.; Xue, J.; Wu, S.; Pei, Y.; Xu, L.; Wang, Y. Cell-Friendly Isolation and PH-Sensitive Controllable Release of Circulating Tumor Cells by Fe3O4@CaCO3 Nanoplatform. Adv. Mater. Interfaces 2021, 8, 1–11, doi:10.1002/admi.202101191.
  7. Popova, V.; Poletaeva, Y.; Pyshnaya, I.; Pyshnyi, D.; Dmitrienko, E. Designing PH-Dependent Systems Based on Nanoscale Calcium Carbonate for the Delivery of an Antitumor Drug. Nanomaterials 2021, 11, 2794.
  8. Please try to increase resolution of Figure 6.

Thank you for the suggestion. We have changed Figure 6 (now Figure 7).

  1. Line 13, “Herein, the core–shell Fe3O4@CaCO3 nanocomposites were successfully obtained by coprecipitation using a magnetic core nanoscale oleic acid-modified Fe3O4, PEG-2000, and Tween 20 as a template for porous CaCO3 coating.” Please re-write this sentence to more clarify.

Thank you for your valuable comments. We have revised the text.

  1. Line 218, “This is probably due to the greater propensity of particles with a high content of magnetite to aggregate during the preparation of samples for TEM.”, please input reference for this issue.

Thank you for your valuable comments. We have inserted the reference. It is probably due to the greater propensity of particles with a high content of magnetite to aggregate during the preparation of samples for TEM [Bondarenko L. et al. RSC advances. 2021].

  1. Also, in other sections of results, there is a lack of thorough discussion on the results. The authors should be more informative and provide more comparison between the results of the current work with former studies.

Thank you for your valuable comments. We have tried to enlarge the discussion of the results. However, only two research groups presented the synthesis of nanoscale Fe3O4@CaCO3 composites as promising multifunctional drug delivery systems [1,2]. There are a few magnetic Fe3O4@CaCO3 hybrids with particles size higher than 1 µm synthesis methods [3-6]. In this way, it is difficult to compare the synthesis result with other works. The aim of the works is not the same.

  1. Vavaev, E.S.; Novoselova, M.; Shchelkunov, N.M.; German, S.; Komlev, A.S.; Mokrousov, M.D.; Zelepukin, I. V.; Burov, A.M.; Khlebtsov, B.N.; Lyubin, E. V.; et al. CaCO3Nanoparticles Coated with Alternating Layers of Poly-L-Arginine Hydrochloride and Fe3O4Nanoparticles as Navigable Drug Carriers and Hyperthermia Agents. ACS Appl. Nano Mater. 2022, 5, 2994–3006, doi:10.1021/acsanm.2c00338.
  2. Wang, P.; Xue, J.; Wu, S.; Pei, Y.; Xu, L.; Wang, Y. Cell-Friendly Isolation and PH-Sensitive Controllable Release of Circulating Tumor Cells by Fe3O4@CaCO3 Nanoplatform. Adv. Mater. Interfaces 2021, 8, 1–11, doi:10.1002/admi.202101191.
  3. Serov, N.; Prilepskii, A.; Sokolov, A.; Vinogradov, V. Synthesis of Plasmin-Loaded Fe3O4@CaCO3 Nanoparticles: Towards Next-Generation Thrombolytic Drugs. ChemNanoMat 2019, 5, 1267–1271, doi:10.1002/cnma.201900359.
  4. Xue, J.; Li, X.; Li, Q.; Lyu, J.; Wang, W.; Zhuang, L.; Xu, Y. Magnetic Drug-Loaded Osteoinductive Fe3O4/CaCO3 Hybrid Microspheres System: Efficient for Sustained Release of Antibiotics. J. Phys. D. Appl. Phys. 2020, 53, doi:10.1088/1361-6463/ab7bb2.
  5. Li, F.H.; Tang, N.; Wang, Y.Q.; Zhang, L.; Du, W.; Xiang, J.; Cheng, P.G. Synthesis and Characterization of Magnetic Carriers Based on Immobilized Enzyme. IOP Conf. Ser. Mater. Sci. Eng. 2018, 359, doi:10.1088/1757-899X/359/1/012044.
  6. Han, P.; Jiang, Z.; Wang, X.; Wang, X.; Zhang, S.; Shi, J.; Wu, H. Facile Preparation of Porous Magnetic Polydopamine Microspheres through an Inverse Replication Strategy for Efficient Enzyme Immobilization. J. Mater. Chem. B 2015, 3, 7194–7202, doi:10.1039/c5tb01094b.
  7. Why did the authors choose the PEG-2000, and Tween 20 in the paper as a template for porous CaCO3 coating?

Thank you for your question. In our past work about nanoscale CaCO3 synthesis, the effect of the composition of the reaction mixture on the characteristics of the resulting calcium carbonate particles was studied.

Popova, V.; Poletaeva, Y.; Pyshnaya, I.; Pyshnyi, D.; Dmitrienko, E. Designing PH-Dependent Systems Based on Nanoscale Calcium Carbonate for the Delivery of an Antitumor Drug. Nanomaterials 2021, 11, 2794.

We have published the effects of adding the following surfactants to the reaction mixture: sodium dodecyl sulfate (anionic detergent), cetyltrimethylammonium bromide (cationic detergent), Tween 20 and Triton X-100 (nonionic detergents), and biocompatible high-molecular-weight compounds (PEG- 1000, -2000, and -6000). Based on DLS data of calcium carbonate particles size obtained in the presence of one or a combination of surfactants it was found that the combination of Tween 20 and PEG-2000 leads to the smallest particle sizes. In the other variants, the particle size exceeded 450 nm.

  1. Table 1 and Table 2, please input size by the TEM images in the Table 1 and Table 2 to compare with size reported by the DLS.

Thank you for your valuable comments. We have revised the text and have presented some discussion.

  1. Please try to more discussion on relationship between size and release studies of drug-loaded nanocomposites.

Thank you for the suggestion. However, the doxorubicin release is not a simple thing. Doxorubicin may interact with other doxorubicin forming various complexes and with nanocomposite surface. Moreover, it is a debatable issue what the size should be chosen: whole of the Fe3O4@CaCO3/DOX or the core, by what method, etc. That is why the correlation between size and release is difficult issue, which can mislead the reader without numerous further investigations.

  1. Does the size and structure of the NP play an important role?It is noteworthy that 100 nm particles or larger generally do not penetrate well throughout the tumour mass, and smaller nanoparticles do not accumulate sufficiently in the tumour vasculature by the enhanced permeability and retention (EPR) effect and do not achieve good tumour penetration.

Thank you for your question. Nanoparticles can accumulate and be retained in tumors from circulating blood due to the effect of EPR. The efficiency of this process can be increased by the application of endogenous stimuli and external stimuli. These, among others, include pH dependence, heating, and magnetic field. For such a path, the most suitable particle size is 100–150 [Cheng X. et al. Frontiers in Molecular Biosciences. 2020]. However, the problem of inefficient penetration of nanoparticles of this size into solid tumors is known due to high extracellular matrix density, increased interstitial fluid pressure, and abnormal vascular structure.

On the contrary, nanoparticles smaller than 20 nm have relatively higher penetrating power and interstitial transport [Huo M. et al. Frontiers in Cell and Developmental Biology. 2021]. However, small nanoplatforms (< 20 nm) are rapidly cleared from the circulating blood via renal filtration, resulting in inefficient tumor accumulation [Bai S. et al. Nano Today. 2021]. To solve this problem, drug delivery systems capable of resizing may be suitable. They must have a large diameter in the bloodstream to achieve passive targeting by the EPR effect and turn into ultra-small particles in tumors with a stimulus to penetrate deep into the tumor [Cheng X. et al. Frontiers in Molecular Biosciences. 2020].

We expect that the two-component nature of the nanocomposite may provide a solution to this problem. When exposed to a low pH tumor area, the carbonate layer will break down, reducing the size of the container. This fact may provide improved diffusion in the tumor. However, we understand that this fact will be highly dependent on the corona protein and needs further research.

  1. Line 336, discussion regarding” 3.5. Human Serum Albumin Interaction with Fe3O4@CaCO3/DOX” is very short. Please do more discussion with supportive literature review.

Thank you for your valuable comments. We have revised the text in section 3.5.

  1. How likely is the new agent approved by FDA?

We don’t clearly understand the reviewer's question. However, the present work is in the initial stage of the investigation. Of course, extensive structure and magnetic properties optimization, biocompatibility studies, in vivo experiments, etc. are required. Despite the promising data demonstrated in our study, there are still many aspects to consider regarding the behavior of the nanocomposite in vitro and in vivo.

  1. It is better to present the data of Fe3O4, CaCO3 in treatment.

We don’t clearly understand your question. However, we have previously published data about DOX-loaded Fe3O4 nanoparticles and DOX-loaded CaCO3 nanoparticles. This work aimed to synthesize DOX-loaded Fe3O4@CaCO3 nanocomposites.

Kovrigina, E.; Chubarov, A.; Dmitrienko, E. High Drug Capacity Doxorubicin-Loaded Iron Oxide Nanocomposites for Cancer Therapy. Magnetochemistry 2022, 8, 54.

Popova, V.; Poletaeva, Y.; Pyshnaya, I.; Pyshnyi, D.; Dmitrienko, E. Designing PH-Dependent Systems Based on Nanoscale Calcium Carbonate for the Delivery of an Antitumor Drug. Nanomaterials 2021, 11, 2794.

  1. In the conclusion, provide a brief explanation about the future perspective of the developed carrier and how it can be modified to exhibit better performance. In the other words, the authors should do the analysis the conclusion section must clearly establish a strong correlation with the proposed topic.

Thank you for your valuable comments. We have revised the text.

  1. Your abstract should clearly state the essence of the problem you are addressing, what you did and what you found and recommend. That will help a prospective reader of the abstract to decide if they wish to read the entire article.

Thank you for your valuable comments. We have revised the text.

  1. Why the authors did not fitted the release data in known release models? 

The major point of the release experiment was to show the pH sensitivity of the experiment. We want to present the difference between the release efficiency at pH 7.4 and acidic media, which provides the prospects of the nanocomposite.

  1. The encapsulation efficiency discussion is very short.

Thank you for your valuable comments. We have revised the text.

Round 2

Reviewer 1 Report

accepted.

Reviewer 2 Report

Recommended for publication as the authors have addressed all the queries. 

Reviewer 3 Report

Authors addressed all comments carefully.